# Can ChatGPT assist authors with abstract writing in medical journals? Evaluating the quality of scientific abstracts generated by ChatGPT and original abstracts

**Taesoon Hwang** [1,2]*, **Nishant Aggarwal**[1,2], **Pir Zarak Khan**[2], **Thomas Roberts**[1], **Amir Mahmood**[1], **Madlen M. Griffiths**[3], **Nick Parsons**[1], **Saboor Khan**[2]

**1** Warwick Medical School, University of Warwick, Coventry, United Kingdom, **2** University Hospitals Coventry and Warwickshire, Coventry, United Kingdom, **3** Royal Devon and Exeter Hospital, Exeter, United Kingdom

* taesoon.hwang@warwick.ac.uk

## Abstract

**Data Availability Statement:** All relevant data are within the paper and its Supporting Information files.

### Introduction

ChatGPT, a sophisticated large language model (LLM), has garnered widespread attention for its ability to mimic human-like communication. As recent studies indicate a potential supportive role of ChatGPT in academic writing, we assessed the LLM's capacity to generate accurate and comprehensive scientific abstracts from published Randomised Controlled Trial (RCT) data, focusing on the adherence to the Consolidated Standards of Reporting Trials for Abstracts (CONSORT-A) statement, in comparison to the original authors' abstracts.

### Methodology

RCTs, identified in a PubMed/MEDLINE search post-September 2021 across various medical disciplines, were subjected to abstract generation via ChatGPT versions 3.5 and 4, following the guidelines of the respective journals. The overall quality score (OQS) of each abstract was determined by the total number of adequately reported components from the 18-item CONSORT-A checklist. Additional outcome measures included percent adherence to each CONOSORT-A item, readability, hallucination rate, and regression analysis of reporting quality determinants.

### Results

Original abstracts achieved a mean OQS of 11.89 (95% CI: 11.23–12.54), outperforming GPT 3.5 (7.89; 95% CI: 7.32–8.46) and GPT 4 (5.18; 95% CI: 4.64–5.71). Compared to GPT 3.5 and 4 outputs, original abstracts were more adherent with 10 and 14 CONSORT-A items, respectively. In blind assessments, GPT 3.5-generated abstracts were deemed most readable in 62.22% of cases which was significantly greater than the original (31.11%; P = 0.003) and GPT 4-generated (6.67%; P<0.001) abstracts. Moreover, ChatGPT 3.5

**Funding:** The author(s) received no specific funding for this work.

**Competing interests:** The authors have declared that no competing interests exist.

exhibited a hallucination rate of 0.03 items per abstract compared to 1.13 by GPT 4. No determinants for improved reporting quality were identified for GPT-generated abstracts.

## Conclusions

While ChatGPT could generate more readable abstracts, their overall quality was inferior to the original abstracts. Yet, its proficiency to concisely relay key information with minimal error holds promise for medical research and warrants further investigations to fully ascertain the LLM's applicability in this domain.

## Introduction

New artificial intelligence (AI) techniques have exploded in sophistication and accessibility in recent times. ChatGPT became the fastest-growing app in history after it was released in November 2022, achieving 100 million active users by January 2023. The utility of AI spans many aspects of modern life which range from automation of daily tasks via virtual assistance to building a disease-specific predictive model supporting cancer diagnosis and even forecasting the emergence of the new SARS-CoV-2 variants [1]. Moreover, in medical research, AI models, like ChatGPT, have shown to comprehensively search and review existing literature, identify knowledge gaps to propose novel hypotheses, and augment data analysis through advanced coding capabilities [2, 3].

ChatGPT is a large language model (LLM) which incorporates machine learning, neural networks and natural language processing (NLP) to emulate human-like conversations and writings. The neural network mimics the neurons of the human brain by which the highly interconnected nodes are arranged in multiple layers and the network processes its input by assigning mathematical representations and weights to different parts of the sentence. This helps to contextualise and assess the relevance of each word in a sentence, and predicts the best possible response to a prompt based on its understanding of the language and its pre-trained database [4]. Recent studies have applied these features of ChatGPT in scientific writing whereby the LLM was able to generate believable abstracts when it was solely provided with a title of an existing study [5]. Furthermore, when it was supplied with a fictional dataset, ChatGPT produced an abstract with appropriate headings, length, and language in addition to correctly conducting statistical analysis and interpretation of its results [6]. These studies have shown ChatGPT to be a valuable writing tool and hinted at its potential to assist scientific writing. However, a comprehensive assessment of its accuracy and reliability remains imperative, and as of this writing, the performance of ChatGPT in this domain has not been objectively examined.

In 2008, the Consolidated Standards of Reporting Trials for abstracts (CONSORT-A)was published with an aim to enhance the overall quality of scientific abstracts, specifically from randomised controlled trials (RCT) [7]. This guideline encompassed essential components that should feature in an abstract and organised them into key categories including the title, trial design, methodology, results, conclusions, trial registration, and funding subsections. This structured approach established a standardised framework for reporting medical abstracts, thereby facilitating a comprehensive and transparent representation of RCTs.

Building on concrete criteria for abstract reporting and the evolving capabilities of AI, the present study aimed to evaluate ChatGPT's potential application in academic writing, with a specific focus on generating scientific abstracts from full RCT reports. The objectives of the

study were to compare the adherence of ChatGPT-generated and original abstracts to the CONSORT-A statement and explore factors that influence the reporting quality. Given its prospective significance in medical research, it was hypothesised that ChatGPT will produce superior abstracts to the original authors.

## Methodology

### Search strategy and study selection

Literature search was conducted on MEDLINE database accessed through PubMed to identify RCTs that were published in journals with the highest impact factors according to the Journal Citation Reports (JCR) 2022 for each medical specialty. This approach was underpinned by our objective to include studies from journals that are not only highly influential but also subject to the rigorous peer-review process. Moreover, the selection of journals was proportionally representative of the medical specialties included in the study. Surgery and medicine, further divided into nine distinct specialities each, collectively represented a substantial portion of the study (~80%), leading to a proportional allocation of journals to these categories. For other specialties, including psychiatry, obstetrics and gynaecology, and paediatrics, the same principle was applied in the journal selection. This balanced approach to the journal selection reinforced the study's comprehensiveness and inclusivity, and thereby enhanced the credibility and generalisability of our findings across various medical disciplines.

The selected journals included the Lancet, New England Journal of Medicine (NEJM), Journal of American Medical Association (JAMA), the British Medical Journal (BMJ), Annals of Surgery, International Journal of Surgery, British Journal of Surgery, the Lancet Child and Adolescent Health, the Lancet Psychiatry and the American Journal of Obstetrics and Gynaecology. To ensure that prior exposure or knowledge of a study did not influence abstract generation by ChatGPT, any RCTs published prior to September 2021 were excluded.

The retrieved studies were stratified into their respective specialties, which included Surgery, Medicine, Paediatrics, Obstetrics and Gynaecology and Psychiatry. Surgery was further categorised into Breast surgery, Cardiothoracic Surgery, ENT, General Surgery, Neurosurgery, Ophthalmology, Orthopaedic Surgery, Urology and Vascular Surgery, meanwhile Medicine was further classified into Cardiology, Endocrinology, Gastroenterology, Haematology, Infectious Diseases, Nephrology, Neurology, Respiratory Medicine and Rheumatology. For each specialty, articles were randomly ordered using Excel (Version 2302, Microsoft, Redmond, Washington) and the first 7±1 studies were selected for inclusion. The retrieved articles underwent further screening whereby feasibility and pilot studies, observational studies, post-trial follow-up studies, and subgroup or secondary analysis of previously reported RCTs were excluded in addition to research protocols, letters/comments to the editor and studies with no full text access.

### Generation of abstract

Both ChatGPT 3.5 and 4 (Version May 24th, 2023, OpenAI, San Francisco, California) were prompted to generate scientific abstracts following an input of full text of a RCT. A new chat was started for each abstract generation to avoid the influence of previous commands and information. The prompt was based on the author guideline of a medical journal in which the RCT was published (see *S1 Fig*). These recommendations encompassed the word count limit and the required subheadings of the abstract. Moreover, after confirming ChatGPT's knowledge of the CONSORT-A, the model was instructed to adhere to the checklist during the abstract generation.

### Primary outcome

Quality of the abstract was determined by assessing its adherence to the 18-item CONSORT-A checklist (*see S2 Fig.*). The checklist was adopted from previous studies which was modified to accommodate all components of the CONSORT-A statement [8, 9]. Each item was given equal weight and graded, dichotomously, 0 or 1, based on its comprehensive and accurate reflection of the main text. It was essential to check the accuracy against the main article to negate 'hallucination', a recognised phenomenon where the LLM produces factually incorrect or misleading information as a result of incomplete data, misinterpretation of its trained dataset or its limited capacity to comprehend the input query. Overall quality score (OQS) was defined by the total number of items that were sufficiently reported and this was presented as number on a scale of 0 to 18 and as percentage of the total number of items.

### Secondary outcomes

Percentage of abstracts in adherence with each of the 18 items of the CONSORT-A checklist was measured. Furthermore, potential predictors of reporting quality in GPT-generated abstracts were investigated which included the word count limit of the abstract, word length of the main report, type of intervention, number of outcome measures and the significance of the study outcome. The readability of the abstracts was evaluated as assessors identified the most comprehensible and clearly presented abstract for each RCT study, and the rate of "hallucination" in output generated by both ChatGPT 3.5 and 4 was also measured.

### Data collection

Assessors were blinded to the abstract allocation, ensuring unbiased evaluation. The abstracts were exclusively generated by TH, who did not partake in the evaluation process. NA, PZK, TR, AM and MMG were involved in appraising the quality of abstracts. Each set consisted of the original, ChatGPT 3.5-generated and ChatGPT 4-generated abstracts and was evaluated by two of the authors. To standardise abstract scoring, the first 5 sets of abstracts were evaluated as a group whereby any misconception of the checklist was clarified. Following this, if there was any disagreement in the abstract assessment, this was discussed and resolved with an independent evaluator, SK.

### Statistical analysis

Continuous data was presented as mean average, proportion (%), simple coefficients and 95% confidence interval, when applicable. Paired T-test was conducted to compare the mean OQS of the original, GPT 3.5- and GPT 4-generated abstracts, meanwhile Pearson's $\chi 2$ test assessed the proportion of abstracts adherent to the CONSORT-A items and readability. Multivariate linear regression analysis was performed to investigate for any determinants associated with higher reporting quality in the GPT 3.5-generated abstracts. Interobserver agreement between the evaluators was assessed using the Cohen κ coefficient whereby a value of greater than 0.6 was considered sufficient. P value of $<0.05$ was considered statistically significant and all tests were conducted via SPSS (Version 28.0, IBM, Armonk, New York) and R (Version 4.3.1, R Core Team, Vienna, Austria) using the R Stats Package (Version 3.6.2).

## Results

*Fig 1* is a flowchart demonstrating the process by which articles were selected for this study. Our search retrieved 722 articles which were published post-September 2021. After the selection process, the articles were subjected to further screening in which 81 articles were excluded

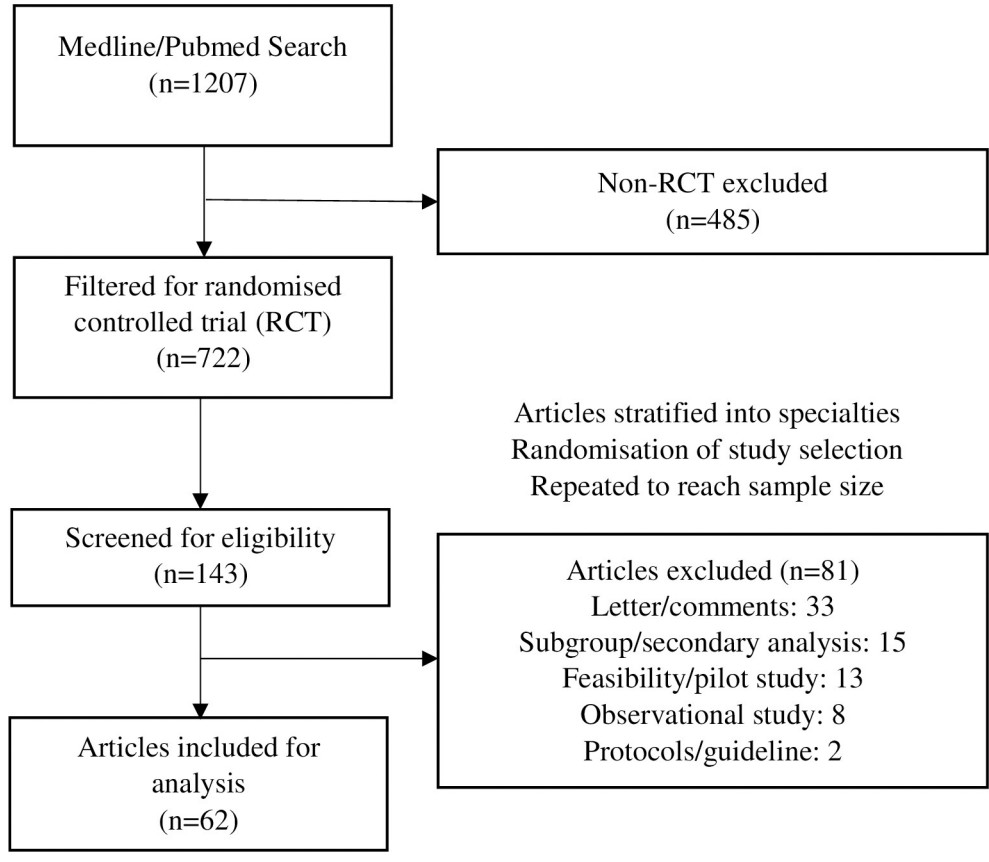

**Fig 1. Flowchart of studies selection.**

in accordance with the study's inclusion criteria. The characteristics of these selected studies are shown on *Table 1*.

## Overall quality scores (OQS)

*Table 2* shows the overall reporting quality across all abstract groups. The original abstracts achieved a mean OQS of 11.89 (95% CI: 11.23–12.54), meanwhile GPT 3.5-generated abstracts scored significantly lower with a mean of 7.89 (95% CI: 7.32–8.46) (P<0.001 95%; CI: 3.16–4.84). GPT 4-generated abstracts demonstrated a mean OQS of 5.18 (95% CI: 4.64–5.71) which was significantly inferior to both the original abstracts (P<0.001; 95% CI: 5.86–7.65) and GPT 3.5-generated abstracts (P<0.00;1 95% CI: 1.99–3.43). Maximum and minimum number of items reported in the original abstract was 17 and 4, respectively, meanwhile GPT 3.5 displayed a narrower range between 12 and 3. GPT 4 demonstrated a maximum score of 11 and recorded a minimum score of 0 as GPT 4 produced 4 abstracts of an unrelated study, thereby falsely reporting all items of the CONSORT-A checklist.

## Reporting of each criterion of CONSORT-A checklist

*Table 3* demonstrates adherence of abstracts to each item of the CONSORT-A checklist. Comparable percentages of abstracts were identified as randomised in the title across all three groups (67.74% vs 67.74% vs 59.68%; original vs GPT 3.5 vs GPT 4, respectively). Trial designs did not differ between original and GPT 3.5, however the criterion was reported in a

**Table 1. Characteristics of selected research articles.**

| Characteristics | Category | N (%) |
|---|---|---|
| Journal | American Journal of Obstetrics and Gynaecology | 5 (8.06%) |
| | Annals of Surgery | 4 (6.45%) |
| | BMJ | 3 (4.84%) |
| | British Journal of Surgery | 1 (1.61%) |
| | International Journal of Surgery | 1 (1.61%) |
| | JAMA | 11 (17.74%) |
| | NEJM | 18 (29.03%) |
| | Lancet | 13 (20.97%) |
| | Lancet Child and Adolescent | 3 (4.83%) |
| | Lancet Psychiatry | 3 (4.83%) |
| Specialty | Surgery [a] | 24 (38.71%) |
| | Medicine [b] | 28 (45.16%) |
| | Paediatrics | 3 (4.84%) |
| | Obstetrics and gynaecology | 4 (6.45%) |
| | Psychiatry | 3 (4.84%) |
| Abstract word count limit | ≤250 | 26 (41.94%) |
| | >250 | 36 (58.06%) |
| Type of intervention | Pharmacological | 36 (58.06%) |
| | Non-pharmacological | 26 (41.94%) |
| Number of outcome measures | ≤5 | 19 (30.65%) |
| | >5 | 43 (69.35%) |
| Significance of results | Significant | 43 (69.35%) |
| | Non-significant | 19 (30.65%) |
| Length of main text | ≤4000 | 19 (30.65%) |
| | >4000 | 43 (69.35%) |

[a] Surgery was further classified into breast surgery, cardiothoracic surgery, ENT, general surgery, neurosurgery, ophthalmology, orthopaedic surgery, urology, and vascular surgery.
[b] Medicine was further classified into cardiology, endocrinology, gastroenterology, haematology, infectious diseases, nephrology, neurology, respiratory medicine, and rheumatology.

significantly higher percentage of original abstracts than the GPT 4 counterparts (41.94% vs 20.97%, respectively; P = 0.02) (see *S1 Table* for statistical comparison between all three groups).

The original abstracts consistently exceeded the GPT-generated abstracts in the majority of the evaluated methodology criteria. Original abstracts significantly out-performed GPT 3.5 and GPT 4 in participants eligibility criteria (75.81% vs 54.84% vs 17.74%, respectively; P = 0.024 original vs GPT 3.5; P<0.001 original vs GPT 4), interventions (85.48% vs 58.06% vs 30.65%, respectively; P = 0.001 original vs GPT 3.5; P<0.001 original vs GPT 4), and method of randomisation (16.13% vs 1.61% vs 0.00%, respectively; P = 0.012 original vs GPT 3.5; P = 0.003 original vs GPT 4). For the remaining criteria including description of study setting, primary outcome measure, allocation concealment, and blinding, GPT 3.5 performed comparably to the original abstracts with one exception of study objectives in which GPT 3.5 demonstrated significantly greater adherence (90.32% vs 69.35%, respectively; P = 0.007).

In comparison to the original abstracts, GPT 4 demonstrated less adherence to further methodology criteria. Original abstracts performed superiorly to GPT 4 abstracts in description of study setting (37.10% vs 8.06%, respectively; P<0.001), primary outcome measure

**Table 2. Overall quality score (OQS) for the original vs. GPT 3.5 vs. GPT 4-generated abstracts.**

|  | Original (%) | GPT 3.5-generated (%) | GPT 4-generated (%) |
|---|---|---|---|
| Mean | 11.89 (66.06%) | 7.89 (43.83%) | 5.18 (28.78%) |
| SD | 2.57 (14.28%) | 2.24 (12.44%) | 2.11 (11.72%) |
| 95% CI | 11.23–12.54 (62.39% - 69.67%) | 7.32–8.46 (40.67% - 47.00%) | 4.64–5.71 (25.78% - 31.72%) |
| Maximum | 17.00 (94.44%) | 12.00 (66.67%) | 11.00 (61.11%) |
| Minimum | 4.00 (22.22%) | 3.00 (16.67%) | 0.00 (0.00%) |

Significance Tests

Original vs. GPT 3.5: P<0.001, 95% CI [3.16–4.84]

Original vs. GPT 4: P<0.001, 95% CI [5.86–7.56]

GPT 3.5 vs. GPT 4: P<0.001, 95% CI [1.99–3.43]

Quality of the abstract was determined by assessing its adherence to the 18-item CONSORT-A (Consolidated Standards of Reporting Trials) checklist (see S1 Fig).

Overall quality score (OQS) was defined by the total number of items that were sufficiently reported and this was presented as number on a scale of 0 to 18 and as a percentage of the total number of items.

Paired T-tests were conducted to compare the mean OQS of the original, GPT 3.5- and GPT 4-generated abstracts

(96.77% vs 56.45%, respectively, P<0.001) and blinding of participants and assessors (59.68% vs 32.26%, respectively, P = 0.004). No difference was found between GPT 4 and original abstracts for the reporting of study objectives and allocation concealment.

**Table 3. Comparison of the adherence of CONSORT-A checklist items by the original abstracts vs GPT 3.5 and GPT 4-generated abstracts.**

| Criterion assessed | Original abstract (N = 62), (%) | GPT 3.5-generated (N = 62), % | GPT 4-generated (N = 62), % |
|---|---|---|---|
| 1. Title | 42 (67.74%) | 42 (67.74%) | 37 (59.68%) |
| 2. Trial design | 26 (41.94%) | 18 (29.03%) | **13 (20.97%)** |
| **METHODOLOGY** | | | |
| 3. Participants, eligibility criteria (a) | 47 (75.81%) | **34 (54.84%)** | **11 (17.74%)** |
| 4. Participants, description of study setting (b) | 23 (37.10%) | 16 (25.81%) | **5 (8.06%)** |
| 5. Interventions | 53 (85.48%) | **36 (58.06%)** | **19 (30.65%)** |
| 6. Objective | 43 (69.35%) | **56 (90.32%)*** | 37 (59.68%) |
| 7. Primary outcome | 60 (96.77%) | 53 (85.48%) | **35 (56.45%)** |
| 8. Randomisation, method (a) | 10 (16.13%) | **1 (1.61%)** | **0 (0.00%)** |
| 9. Randomisation, allocation concealment (b) | 3 (4.84%) | 0 (0.00%) | 0 (0.00%) |
| 10. Blinding | 37 (59.68%) | 33 (53.23%) | **20 (32.26%)** |
| **RESULTS** | | | |
| 11. Numbers randomised | 51 (82.26%) | **19 (30.65%)** | **11 (17.74%)** |
| 12. Numbers analysed | 32 (51.61%) | **2 (3.23%)** | **2 (3.23%)** |
| 13. Outcome, results (a) | 59 (95.16%) | **28 (45.16%)** | **12 (19.35%)** |
| 14. Outcome, effects size and precision (b) | 58 (93.55%) | **26 (41.94%)** | **10 (16.13%)** |
| 15. Harms | 41 (66.13%) | **9 (14.52%)** | **8 (12.90%)** |
| 16. Conclusions | 61 (98.39%) | 62 (100.00%) | 58 (93.55%) |
| 17. Trial registration | 55 (88.71) | **37 (59.68%)** | **26 (41.94%)** |
| 18. Funding | 36 (58.06%) | **19 (30.65%)** | **18 (29.03%)** |

**Emboldened** values represent significant difference in comparison to the original abstracts for the corresponding CONSORT (Consolidated Standards of Reporting Trials)-A criterion.

* GPT 3.5 scored significantly higher than the original abstracts.

Pearson's $\chi^2$ test was performed to compare the adherence of each CONSORT-A item.

When compared to GPT 4, GPT 3.5-generated abstracts demonstrated significantly greater adherence than GPT 4-generated counterpart in the reporting of eligibility criteria (P<0.001), description of study setting (P = 0.017), intervention intended (P = 0.004), study objective and hypothesis (P<0.001), primary outcome measure (P = 0.001) and blinding (P = 0.029) (see *S1 Table*).

In terms of adherence to the results criteria, the original abstract significantly outperformed both GPT 3.5 and 4 across all assessed items. This included reporting of participants randomised to each group (82.26% vs 30.65% vs 17.74%, respectively; P<0.001 original vs GPT 3.5; P<0.001 original vs GPT 4), number of participants analysed in each group (51.61% vs 3.23% vs 3.23%, respectively; P<0.001 original vs GPT 3.5; P<0.001 original vs GPT 4), primary outcome results for each group (95.16% vs 45.16% vs 19.35%, respectively; P<0.001 original vs GPT 3.5; P<0.001 original vs GPT 4), effect size and precision of primary outcome results (93.55% vs 41.94% vs 16.13%, respectively; P<0.001 original vs GPT 3.5; P<0.001 original vs GPT 4), and the adverse effects of each group (66.13% vs 14.52% vs 12.90%, respectively; P<0.001 original vs GPT 3.5; P<0.001 original vs GPT 4). Meanwhile, GPT 3.5-generated abstracts performed superiorly to GPT 4 counterparts in the primary outcome results (P = 0.004), and effect size and precision(P = 0.003).

The trial conclusions were reported comparably between all three abstract groups (98.39% vs 100.00% vs 93.55%; original vs GPT 3.5 vs GPT 4, respectively). The original abstracts significantly outperformed GPT 3.5 and GPT 4 in reporting trial registration (88.71% vs 59.68% vs 41.94%, respectively; P<0.001 original vs GPT 3.5; P<0.001 original vs GPT 4) and source of funding (58.06% vs 30.65% vs 29.03%, respectively; P = 0.004 original vs GPT 3.5; P = 0.002 original vs GPT 4).

## Analysis of variables associated with the OQS of GPT 3.5-generated abstracts

*Table 4* displays the results of multivariable linear regression analysis for GPT 3.5-generated abstracts. Word count, type of intervention, number of outcome measures, significance of trial results and the length of main report had no significant association with overall quality score of the GPT 3.5-generaated abstracts.

## Readability of abstracts

In blinded assessment, abstracts generated by ChatGPT 3.5 demonstrated superior readability in comparison to the original and GPT 4 (62.22% vs 31.11% vs 6.67%, respectively; P = 0.003

**Table 4. Multivariable linear regression analysis of variables associated with the Overall Quality Score of GPT 3.5-generated abstracts.**

| Characteristics | Category | Multivariable analysis | | |
|---|---|---|---|---|
| | | Coefficient | 95% CI | P-value |
| Word count | ≤250 | 1 | - | |
| | >250 | 0.99 | -0.17–2.15 | 0.094 |
| Type of intervention | Pharmacological | 1 | - | |
| | Non-pharmacological | 0.98 | -0.18–2.14 | 0.097 |
| Number of outcome measures | ≤5 | 1 | - | |
| | >5 | 0.054 | -1.22–1.33 | 0.932 |
| Significance of results | Significant | 1 | - | |
| | Non-significant | 0.011 | -1.27–1.29 | 0.986 |
| Length of main report | ≤4000 | 1 | - | |
| | >4000 | -0.89 | -2.18–0.40 | 0.171 |

**Table 5. Blinded assessment of the readability of ChatGPT-generated vs original abstracts.**

| | Number of abstracts selected as most readable, N (%) |
|---|---|
| Original | 14 (31.11%) |
| GPT 3.5-generated | 28 (62.22%) |
| GPT 4-generated | 3 (6.67%) |

Significance Tests

Original vs GPT 3.5, P = 0.003

Original vs GPT 4, P = 0.003

GPT 3.5 vs GPT 4, P<0.001

It should be noted that the readability data was only available for 45 out of the 62 abstracts initially selected for this study due to incomplete evaluations in the assessment process.

Pearson's χ2 test was used to compare the performance of each abstract subgroup.

GPT 3.5 vs original; P<0.001 GPT 3.5 vs GPT 4) (*Table 5*). However, it should be noted that the readability data was available for 45 out of the 62 abstracts initially selected for this study due to incomplete evaluations in the assessment process.

## Hallucination in ChatGPT-generated abstracts

ChatGPT 4 hallucinated on a mean average of 1.13 items per abstract, meanwhile GPT 3.5 demonstrated noticeably lower rate of hallucination at 0.03 items per abstract. Specifically, GPT 3.5 hallucinated on two occasions whereby it inaccurately reported the number of participating centres and misrepresented the duration for which the primary outcome was assessed. Excluding the four instances in which GPT 4 generated irrelevant abstracts, the LLM conflated secondary and primary outcome measures in three different abstracts.

## Interobserver agreement

Average Cohen κ coefficient for all assessed criteria was > 0.6, indicating a substantial agreement between our evaluators (see Table 6).

## Discussion

This exploratory study aimed to objectively assess the current capacity of ChatGPT in medical research, specifically in generating accurate and comprehensive scientific abstracts across multiple fields of biomedical research. When prompted, ChatGPT generated an authentic-looking abstract with an appropriate structure and concise language while it attempted to extract relevant details to the methodology and results components of a RCT report. However, in comparison to the original abstracts, GPT-generated abstracts demonstrated significantly inferior overall quality as the original abstracts outperformed GPT 3.5 and GPT 4 by 22.22% and 37.30% in the OQS, respectively. Moreover, the original abstracts outperformed GPT 3.5 and GPT 4 in 10 and 14 of the 18 items from the CONSORT-A checklist, respectively, meanwhile no discernible association was identified between the evaluated study characteristics and the overall quality of GPT-generated abstracts. However, abstracts generated by GPT 3.5 were deemed to be most readable in 62.22% of cases in comparison to the GPT 4 and original counterparts, and it demonstrated minimal hallucination rate of 0.03 errors per abstract.

Several factors may account for ChatGPT's relative underperformance in its abstract generation. Firstly, the comparator original abstracts were from studies published in high impact journals which are known for rigorous peer review. The mean OQS of these abstracts was

**Table 6. Interobserver reliability in the assessment of medical abstracts.**

| Criterion assessed | Pairs of Evaluators | | | |
| --- | --- | --- | --- | --- |
| | ZPK & TR | TR & AM | NA & MMG | Average Kappa point |
| 1. Title | **0.942** | **1.000** | **1.000** | **0.981** |
| 2. Trial design | **0.690** | **0.745** | **0.931** | **0.789** |
| **METHODOLOGY** | | | | |
| 3. Participants, eligibility criteria (a) | **0.692** | **0.824** | **0.774** | **0.763** |
| 4. Participants, description of study setting (b) | **0.667** | **0.778** | **0.757** | **0.734** |
| 5. Interventions | **0.775** | 0.577 | **0.728** | **0.693** |
| 6. Objective | 0.299 | **0.895** | **0.785** | **0.660** |
| 7. Primary outcome | **0.697** | 0.584 | **0.658** | **0.646** |
| 8. Randomisation, method (a) | **1.000** | **1.000** | **0.882** | **0.961** |
| 9. Randomisation, allocation concealment (b) | 0.378 | **1.000** | **1.000** | **0.793** |
| 10. Blinding | **0.822** | **0.958** | **0.975** | **0.918** |
| **RESULTS** | | | | |
| 11. Numbers randomised | **0.955** | **0.956** | **0.900** | **0.937** |
| 12. Numbers analysed | **0.834** | **1.000** | **0.902** | **0.912** |
| 13. Outcome, results (a) | **0.822** | **0.750** | **0.926** | **0.833** |
| 14. Outcome, effects size and precision (b) | **0.911** | **0.621** | **0.950** | **0.827** |
| 15. Harms | **0.724** | **0.702** | **0.642** | **0.689** |
| 16. Conclusions | **1.000** | 0.484 | 0.491 | **0.658** |
| 17. Trial registration | **1.000** | **0.787** | **1.000** | **0.929** |
| 18. Funding | **0.945** | **0.833** | **1.000** | **0.926** |

**Emboldened** values represent substantial agreement of kappa point > 0.6

66.06%, which is considerably higher than the scores, ranging between 32.6% and 54.1%, typically reported from a broader collection of journals [8–12]. Hence, with an average score of 43.83%, GPT 3.5 may have performed comparatively against a more diverse selection of abstracts. Moreover, current literature demonstrates a varying degree of adherence to the CONSORT-A guidelines [13]. Given that ChatGPT is trained on vast dataset, inclusive of less adherent abstracts, it is possible for the LLM to have inferred that strict adherence to the CONSORT guideline is not always necessary. This study also did not exhaustively explore all possible prompt options and we intentionally used a basic prompt for uniformity as the purpose of this study was to discern the strengths and weaknesses of the LLM, rather than to optimise prompt engineering. Moreover, repeated modification of prompts for each abstract generation would have introduced inter-abstract variability and consequently undermine the robustness of our assessment.

A distinctive strength of the ChatGPT abstracts was its ability to present research findings concisely in easy-to-understand terms as GPT 3.5-generated abstracts were selected as most readable in 62.22% of the assessed studies. A noteworthy observation was the apparent inverse relationship between readability and scoring on our checklist. While GPT-generated abstracts adeptly summarised studies and communicated their implications, they frequently lacked the specific details to score favourably on our checklist. Although direct measurement of readability was beyond the scope of this study, our findings were corroborated by Eppler et al. in which GPT-generated texts outperformed original abstracts in the Global Readability Score, Flesch Kincade Reading Ease and other various readability metrics [14]. Therefore, AI could serve as an invaluable tool in translating complex scientific texts into more accessible versions, hence promoting higher level of comprehension and engagement from the general public.

In addition, ChatGPT demonstrated competence in accurately comprehending and extracting the objectives and the conclusions of the provided studies. ChatGPT 3.5 identified the aims and objectives more consistently than the original abstracts and correctly reported the conclusion in all its generated abstracts. Moreover, ChatGPT 3.5 made very few mistakes with a hallucination rate of 0.03 items per abstract, which outperformed the expectations based on recent literature. For example, when tasked with formulating research protocols, ChatGPT generated false references at a rate of 54.5%, and when the LLM was prompted to generate a full scientific case report, it presented inaccurate differentials despite receiving comprehensive key information [15, 16]. It should be noted that these studies assessed "hallucination" within the framework of content generation, meanwhile our study highlighted ChatGPT's capability to extract key information from a provided text, which possibly explains the observed discrepancy. Nevertheless, the capacity to produce key insights of a given topic holds significant potential for scientific research, particularly in literature reviews, and offers avenues in which existing AI research assistants can be further enhanced to promptly collate up-to-date knowledge of a specific research topic. Moreover, ChatGPT's aptitude for summarisation is invaluable for abstract writing as it lays down a foundational blueprint for authors to subsequently refine. The LLM could optimise word count in adjustment to the journal's requirement, provide alternative ways of phrasing texts to enhance reader engagement and support non-native English speakers with drafting scientific abstracts, thereby broadening the global participation of biomedical researchers. Collectively, these attributes of ChatGPT position the LLM as a promising tool, not only for abstract generation but also as a broader asset for academic writing.

In this study, both models of ChatGPT were evaluated in which GPT 3.5, the preceding model, generated abstracts of superior quality than GPT 4. It outperformed its successor by 15.06% in the OQS and tended to create more accurate abstracts with fewer "hallucinations". The underlying reason for these discrepancies remains unclear as OpenAI is yet to disclose explicit details to the parameters, architecture nor the hardware used for training GPT 4. However, our finding was reinforced in a study from Stanford University which evaluated both models of ChatGPT across diverse domains, encompassing code generation, visual reasoning, handling of sensitive questions and mathematical problem-solving. In particular, when prompted to answer maths problems, GPT 3.5 demonstrated 86.8% accuracy in comparison to 2.4% shown by GPT 4, and this was attributed to GPT 3.5's superiority in breaking down complex problems into smaller intermediate steps, a phenomenon known as the "chain of thoughts" effects [17]. Parallel to our study, it is plausible that GPT 3.5's aptitude for deconstructing complex commands resulted in superior abstract generation compared to its successor. In addition to this, the assumed broader training database and variance in fine-tuning approaches in GPT 4 could have further compromised its capacity for specialised tasks such as those required in this study.

Despite the strengths of ChatGPT in scientific writing, it is imperative to recognise the associated risks and pitfalls. Firstly, LLMs draw from expansive dataset, which could unintentionally reflect biases related to sex, ethnicity, and language. Given that the training data for these LLMs predominantly originate from well-funded institutions in affluent, English-speaking countries, there exists a risk for the underrepresentation of minority groups [18]. Secondly, while the proficiency of LLMs to generate credible information is commending, it can sometimes be misleading. For instance, when prompted to generate a literature review, ChatGPT provided superficial summary that was far from the knowledge of an expert in the field and it faltered in providing correct references, meanwhile in our study, GPT 4 generated 4 abstracts which were entirely unrelated to the given topic [19, 20]. This propensity to generate plausible yet fictitious content intertwines with issues of plagiarism. LLMs could inadvertently extract

and apply content from its training data into their generated output which poses a real challenge to trust in medical research, especially as the difference between author written and LLM-generated texts gradually narrows. Amidst these complexities, the need for transparency becomes paramount and to preserve the integrity of medical research, authors must diligently acknowledge the use of AI tools and uphold accountability in verifying LLM-generated content.

## Strengths and limitations

There are many strengths to this study which includes diverse selection of studies across various specialties and exclusion of studies published prior to September 2021 to minimise the influence of existing knowledge base in the abstract generation. Moreover, this study stands as a pioneer in utilising an objective checklist for evaluating scientific text generated by the LLM. However, there are certain limitations to consider and as aforementioned, the comparator original abstracts exhibited a higher overall CONOSRT-A score than typically observed, hence potentially underestimating ChatGPT's capacity for abstract generation. Furthermore, the findings of this study primarily pertain to the May 24 version of ChatGPT and may not reflect on the current capacity of ChatGPT as the LLM is continuously fine-tuned by the engineers with the latest update rolled out on 25th September 2023. This study also did not distinguish between hallucinated and unreported items though drawing such distinction could have been insightful. However, the authors of this study agreed that misleading content, whether false or omitted, equally compromises the quality of the abstract and hence a dichotomous scoring system was adopted. Lastly, this study has only assessed GPT's ability to summarise large text into abstracts, not its capacity for generating original content, therefore cautions must be made in extrapolating our study findings to the broader scientific writing.

## Conclusions

ChatGPT performed inferiorly to the authors in generating scientific abstracts of CONSORT-A standards. However, it is commendable that ChatGPT produced authentic-looking abstracts with an appropriate structure and summary of the main report with minimal guidance and error. Given the exploratory nature of this study, definitive conclusion regarding GPT's efficacy and applicability in abstract writing remain yet to be established. Therefore, further investigations employing objective evaluation measures will be imperative to ascertain the true relevance and potential of ChatGPT in academic writing and medical research.

## Supporting information

**S1 Fig. Prompt for abstract generation and ChatGPT response.**
(DOCX)

**S2 Fig. CONSORT-A checklist.**
(DOCX)

**S1 Table. Comparison of the adherence of CONSORT-A checklist items by the original abstracts vs GPT 3.5 vs GPT 4-generated abstracts.**
(DOCX)

## Author Contributions

**Conceptualization:** Taesoon Hwang, Pir Zarak Khan, Thomas Roberts, Amir Mahmood.

**Data curation:** Nishant Aggarwal, Pir Zarak Khan, Thomas Roberts, Amir Mahmood, Madlen M. Griffiths.

**Formal analysis:** Taesoon Hwang, Nick Parsons.

**Methodology:** Taesoon Hwang, Pir Zarak Khan, Thomas Roberts, Amir Mahmood.

**Supervision:** Saboor Khan.

**Writing – original draft:** Taesoon Hwang, Nishant Aggarwal.

**Writing – review & editing:** Taesoon Hwang, Nishant Aggarwal, Saboor Khan.

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
