## [Decision Letter · Decision Letter 0]

16 Nov 2023

PONE-D-23-33840Can ChatGPT Assist Authors with Abstract Writing? Evaluating the Quality of Scientific Abstracts Generated by ChatGPT and Original AbstractsPLOS ONE

Dear Dr. Hwang,

Thank you for submitting your manuscript to PLOS ONE. After careful consideration, we feel that it has merit but does not fully meet PLOS ONE’s publication criteria as it currently stands. Therefore, we invite you to submit a revised version of the manuscript that addresses the points raised during the review process.

We look forward to receiving your revised manuscript.

Kind regards,

Takayuki Mizuno, Ph. D.

Academic Editor

PLOS ONE

Journal Requirements:

Additional Editor Comments:

Please follow the comments of referees 1 and 2 and brush up your manuscript so that it can be easily understood by a wide range of readers.

Reviewers' comments:

Reviewer's Responses to Questions

**Comments to the Author**

1. Is the manuscript technically sound, and do the data support the conclusions?

Reviewer #1: Yes

Reviewer #2: Yes

2. Has the statistical analysis been performed appropriately and rigorously? 

Reviewer #1: Yes

Reviewer #2: Yes

3. Have the authors made all data underlying the findings in their manuscript fully available?

Reviewer #1: Yes

Reviewer #2: Yes

4. Is the manuscript presented in an intelligible fashion and written in standard English?

Reviewer #1: Yes

Reviewer #2: Yes

5. Review Comments to the Author

Reviewer #1: This is an interesting study. you may modify the flow chart as the current one is more vertically oriented. To be specific, PubMed and Medline are two different terms, one is a search engine and one is a bibliographic database.

Reviewer #2: With a recent surge of application of language learning models in clinical medicine and dentistry, I find this study timely and relevant. The study is methodologically sounds, however, I have few minor suggestions that I have highlighted below:

1. Line 1: Since thee topic is focussed on medical journals, I suggest to modify the title to "Can ChatGPT Assist Authors with Abstract Writing in Medical Journals? Evaluating the Quality of Scientific Abstracts Generated by ChatGPT and Original Abstracts"

2. Line 79: Rephrase the line. Equator is a database and it did not introduce CONSORT-A. It is an independent research group.

3. Line 95: Elaborate on how journals were selected?

4. Line 130: Explain the term "hallucination"

5. Line 143: I see the authors are trained and calibrated. Include reliability scores for the examiners

6. Table 1: Although this topic is specific to general journals, In addition to general medicine and surgery, I see more emphasis on three specialties: pediatrics, O&G, and Psychiatry. Kindly explain the reason for this focus.

6. PLOS authors have the option to publish the peer review history of their article (what does this mean?). If published, this will include your full peer review and any attached files.

Reviewer #1: No

Reviewer #2: **Yes: **Jayakumar Jayaraman

---

## [Author Response · Author response to Decision Letter 0]

29 Dec 2023

Dear Editor,

We would like to thank the editor and reviewers for their detailed feedback on our manuscript. We have addressed all comments raised by the reviewers and responded with detailed point-by-point explanations. Marked-up copy of the manuscript highlights the changes made in response to the reviewer’s comments, and all page and line numbers mentioned below refer to the marked-up copy. We hope that the changes made to our manuscript adequately address the reviewers’ comments and hence allow for the publication of our novel study. 

Yours sincerely,

Taesoon Hwang

Reviewer’s Comments

Reviewer #1: This is an interesting study. you may modify the flow chart as the current one is more vertically oriented. To be specific, PubMed and Medline are two different terms, one is a search engine and one is a bibliographic database

We thank the reviewer for their comment on the figure. We have edited the presentation of the flowchart in Figure 1 and have clarified that Medline search was conducted through the PubMed platform in Line 96.

Reviewer #2: With a recent surge of application of language learning models in clinical medicine and dentistry, I find this study timely and relevant. The study is methodologically sounds, however, I have few minor suggestions that I have highlighted below:

1. Line 1: Since thee topic is focussed on medical journals, I suggest to modify the title to "Can ChatGPT Assist Authors with Abstract Writing in Medical Journals? Evaluating the Quality of Scientific Abstracts Generated by ChatGPT and Original Abstracts"

We appreciate the reviewer’s suggestion to modify the title of our submitted work. We agree that specifying “Can ChatGPT Assist Authors with Abstract Writing in Medical Journals?” better captures the essence of our work and clarifies the context in which the abstracts were evaluated.

2. Line 79: Rephrase the line. Equator is a database and it did not introduce CONSORT-A. It is an independent research group.

We thank the reviewer for the correction. As suggested, we have addressed this point in Lines 77-86.

3. Line 95: Elaborate on how journals were selected?

We appreciate the reviewer’s comment on elaborating the journal selection process. We agree with this point and have included a detailed explanation in Lines 96-114. 

4. Line 130: Explain the term "hallucination"

We thank the reviewer for this point and appreciate that “hallucination” can be an unfamiliar term for new readers. We have clarified the definition of hallucination in Lines 149-152.

5. Line 143: I see the authors are trained and calibrated. Include reliability scores for the examiners

We appreciate the reviewer’s point and recognise the need to demonstrate reliability in the assessment of abstracts between our evaluators. We have included an additional subsection highlighting a substantial agreement between our evaluators in Lines 278-281 and we have included a table to illustrate this (see Table 6). 

6. Table 1: Although this topic is specific to general journals, In addition to general medicine and surgery, I see more emphasis on three specialties: pediatrics, O&G, and Psychiatry. Kindly explain the reason for this focus.

We thank the reviewer for this comment. In this study, our overarching goal was to assess ChatGPT’s capabilities across a broad spectrum of medical specialties. While general medicine and surgery are core areas of focus, we also recognised the importance of including other pivotal specialties such as paediatrics, obstetrics and gynaecology and psychiatry. These fields represent distinct and critical aspects of medical science and by incorporating them, we sought to provide a more comprehensive evaluation of ChatGPT’s utility in various medical contexts. 

The table’s apparent emphasis on paediatrics, obstetrics and gynaecology, and psychiatry should not be interpreted as an additional focus on these fields. Instead, it demonstrates our deliberate effort to ensure that the study reflects the broad spectrum of medical disciplines. For ease of presentation and to improve the table’s readability, we grouped a multitude of disciplines under the umbrella terms ‘Surgery’ and ‘Medicine’. These broad categories are further delineated into nine sub-specialties each as represented by the superscripts a and b. This structuring choice was made purely for presentation purposes. It is important to clarify that every specialty whether it falls under the broad categories or is one of the three specifically mentioned fields, holds equal value to our study and this approach highlights the comprehensive nature of our analysis meanwhile it reinforces the applicability of our findings across the diverse medical fields. We have explained the rationale for including paediatrics, O&G, and psychiatry in Lines 105-114.

---

## [Decision Letter · Decision Letter 1]

11 Jan 2024

Can ChatGPT Assist Authors with Abstract Writing in Medical Journals? Evaluating the Quality of Scientific Abstracts Generated by ChatGPT and Original Abstracts

PONE-D-23-33840R1

Dear Dr. Hwang,

We’re pleased to inform you that your manuscript has been judged scientifically suitable for publication and will be formally accepted for publication once it meets all outstanding technical requirements.

Kind regards,

Takayuki Mizuno, Ph. D.

Academic Editor

PLOS ONE

Additional Editor Comments (optional):

Reviewers' comments:

Reviewer's Responses to Questions

**Comments to the Author**

1. If the authors have adequately addressed your comments raised in a previous round of review and you feel that this manuscript is now acceptable for publication, you may indicate that here to bypass the “Comments to the Author” section, enter your conflict of interest statement in the “Confidential to Editor” section, and submit your "Accept" recommendation.

Reviewer #1: All comments have been addressed

Reviewer #2: All comments have been addressed

2. Is the manuscript technically sound, and do the data support the conclusions?

Reviewer #1: Yes

Reviewer #2: Yes

3. Has the statistical analysis been performed appropriately and rigorously? 

Reviewer #1: Yes

Reviewer #2: Yes

4. Have the authors made all data underlying the findings in their manuscript fully available?

Reviewer #1: Yes

Reviewer #2: Yes

5. Is the manuscript presented in an intelligible fashion and written in standard English?

Reviewer #1: Yes

Reviewer #2: Yes

6. Review Comments to the Author

Reviewer #1: The title needs mention of the GPT version as free and paid versions are there and their capabilities are different in terms of data processing and responding. In methodology also it should be mentioned.

Reviewer #2: The authors have adequately addressed the comments raised in the previous review. I do not have anything to add.

7. PLOS authors have the option to publish the peer review history of their article (what does this mean?). If published, this will include your full peer review and any attached files.

Reviewer #1: No

Reviewer #2: **Yes: **Jayakumar Jayaraman

---

## [Editor Report · Acceptance letter]

23 Jan 2024

PONE-D-23-33840R1 

PLOS ONE

Dear Dr. Hwang, 

I'm pleased to inform you that your manuscript has been deemed suitable for publication in PLOS ONE. Congratulations! Your manuscript is now being handed over to our production team.

Kind regards, 

on behalf of

Dr. Takayuki Mizuno 

Academic Editor

PLOS ONE